# Ceritinib-Induced Regression of an Insulin-Like Growth Factor-Driven Neuroepithelial Brain Tumor

**DOI:** 10.3390/ijms20174267

**Published:** 2019-08-30

**Authors:** Alexandra Russo, Claudia Paret, Francesca Alt, Jürgen Burhenne, Margaux Fresnais, Wolfgang Wagner, Martin Glaser, Hannah Bender, Sabrina Huprich, Patrick N. Harter, Katharina Filipski, Nadine Lehmann, Nora Backes, Lea Roth, Larissa Seidmann, Clemens Sommer, Marc A. Brockmann, Torsten Pietsch, Marie A. Neu, Arthur Wingerter, Jörg Faber

**Affiliations:** 1Pediatric Hematology/Oncology, Children’s Hospital, University Medical Center of the Johannes Gutenberg-University Mainz, 55131 Mainz, Germany; 2University Cancer Center of the University Medical Center of the Johannes Gutenberg-University, 55131 Mainz, Germany; 3German Cancer Consortium (DKTK), site Frankfurt/Mainz, Germany, German Cancer Research Center (DKFZ), 69120 Heidelberg, Germany; 4Department of Clinical Pharmacology and Pharmacoepidemiology, Heidelberg University, 69120 Heidelberg, Germany; 5German Cancer Consortium (DKTK)-German Cancer Research Center (DKFZ), 69120 Heidelberg, Germany; 6Section of Pediatric Neurosurgery, Department of Neurosurgery, University Medical Center of the Johannes Gutenberg University Mainz, 55131 Mainz, Germany; 7Neurological Institute (Edinger-Institute), Goethe-University Medical School, 60528 Frankfurt am Main, Germany; 8Frankfurt Cancer Institute (FCI), 60596 Frankfurt am Main, Germany; 9Institute of Pathology, University Medical Center of the Johannes Gutenberg University Mainz, 55131 Mainz, Germany; 10Institute of Neuropathology, University Medical Center of the Johannes Gutenberg University Mainz, 55131 Mainz, Germany; 11Department of Neuroradiology, University Medical Center of the Johannes Gutenberg University, 55131 Mainz, Germany; 12Department of Neuropathology, DGNN Brain Tumor Reference Center, University of Bonn, 53127 Bonn, Germany

**Keywords:** ATO, ceritinib, IGF, SHH, NOTCH1, WNT

## Abstract

The insulin-like growth factor (IGF) pathway plays an important role in several brain tumor entities. However, the lack of inhibitors crossing the blood–brain barrier remains a significant obstacle for clinical translation. Here, we targeted the IGF pathway using ceritinib, an off-target inhibitor of the IGF1 receptor (IGF1R) and insulin receptor (INSR), in a pediatric patient with an unclassified brain tumor and a *notch receptor 1* (*NOTCH1*) germline mutation. Pathway analysis of the tumor revealed activation of the sonic hedgehog (SHH), the wingless and integrated-1 (WNT), the IGF, and the Notch pathway. The proliferation of the patient tumor cells (225ZL) was inhibited by arsenic trioxide (ATO), which is an inhibitor of the SHH pathway, by linsitinib, which is an inhibitor of IGF1R and INSR, and by ceritinib. 225ZL expressed INSR but not IGF1R at the protein level, and ceritinib blocked the phosphorylation of INSR. Our first personalized treatment included ATO, but because of side effects, we switched to ceritinib. After 46 days, we achieved a concentration of 1.70 µM of ceritinib in the plasma, and after 58 days, MRI confirmed that there was a response to the treatment. Ceritinib accumulated in the tumor at a concentration of 2.72 µM. Our data suggest ceritinib as a promising drug for the treatment of IGF-driven brain tumors.

## 1. Introduction

Brain tumors are the most common solid tumors and a leading cause of cancer-related death in children. The advances in molecular biology have identified critical cellular changes within pediatric brain tumors, suggesting that molecularly targeted therapy may improve the treatment of these patients.

The insulin-like growth factor (IGF) pathway has long been recognized for its role in tumorigenesis and growth. The IGF axis includes three ligands (IGF1, IGF2, and insulin) and two homolog receptors, insulin like growth factor 1 receptor (IGF1R) and insulin receptor (INSR; for a review, see [1]). In the brain, IGF1R, IGF1, and IGF2 are highly expressed during embryonic and early postnatal development, and decrease substantially during adolescence [2]. In brain tumors, the IGF axis is frequently activated in medulloblastoma [3,4] glioblastoma [5], and ependymoma [6]. Several strategies have been tested to interfere with IGF signaling, including IGF1R blockade by monoclonal antibodies, small molecule tyrosine kinase inhibitors of IGF1R and INSR, and ligand neutralizing strategies [7], but so far, there are no US Food and Drug Administration (FDA)-approved drugs available. Thus, drug repositioning i.e., the strategy of using existing drugs originally developed for one disease to treat other indications, may represent a way forward to rapid clinical use.

Ceritinib gained US Food and Drug Administration approval in 2014 for the treatment of patients with ALK-positive metastatic non-small-cell lung cancer (NSCLC) who have progressed on or are intolerant to crizotinib [8]. Ceritinib can also inhibit INSR and IGF1R [9], suggesting a potential role in the development of therapy protocols targeting the IGF axis. Ceritinib indeed has been used to successfully inhibit IGF1R phosphorylation in models of rhabdomyosarcoma [10] and of high-grade neuroepithelial tumor with BCOR alteration (HGNET-BCOR) [11].

Here, we targeted the IGF pathway in a pediatric patient with a neuroepithelial tumor that was neuropathologically classified as an anaplastic ependymoma, but could not be assigned to a defined DNA-methylation class of brain tumors. Our in vitro and in vivo data indicate that ceritinib can be used to target INSR, and that ceritinib penetrates the human blood–brain barrier (BBB).

## 2. Results

### 2.1. Clinical Description

A 4-year-old female patient without any relevant prior medical condition was transferred to our center after the gross total microsurgical resection of a 5.1 × 3.6 × 2.1 cm hemorrhagic tumor in the right parieto-occipital lobe. The postoperative staging scans revealed no metastases. The tumor was classified as an anaplastic ependymoma (WHO III), and the patient received focal irradiation according to the international HIT-MED registry (Version 1.0, 12 January 2014). Five months after cessation of radiotherapy, a first metastatic relapse occurred in the temporofrontal region. Gross total microsurgical resection of the first metastasis was performed, and we initiated systemic chemotherapy based on cycles of cyclophosphamide/vincristine and carboplatin/etoposide according to the HIT-MED Guidance (Ependymoma, M0, R0, 4–21 years and protocol version 3.0, 16th September 2015). Four months later, the patient developed a second metastasis/metachronous tumor in the temporomesial region, and we decided to switch to a personalized treatment based on the molecular analysis of the tumor.

### 2.2. Histopathology

The histopathological report of the primary tumor was suggestive of a small, round, blue cell tumor with co-expression of neuronal antigens (NeuN and CD56) and epithelial antigens (EMA) (Figure 1B–D). Ki67 staining was indicative of more than 10% proliferating tumor cells (Figure 1E). Further central reference pathological review and additional molecular evaluation (RT-PCR and sequencing) led to the diagnosis of an anaplastic ependymoma (WHO III) carrying a *C11orf95-RELA* fusion. However, L1Cam, a surrogate marker for *RELA* fusion-gene positivity, was detected only in a subfraction of tumor cells (Figure 1F). Moreover, in this specimen, subpopulations of cells showed nuclear accumulation of the p65RelA protein as specific indication of NFκB activation [12] (not shown). The tumor protein P53 (TP53) gene product was accumulated in approximately 20% of cells. Histology of the first metastatic relapse revealed a tumor lacking any characteristic patterns (Appendix A). The tumor cells were round to oval and of varying size, with only a scant cytoplasm. Sometimes, distinct nucleoli were present. Mitotic activity was brisk. There was geographic necrosis. Immunohistochemically, most tumor cells were positive for EMA, while some displayed positivity for L1Cam. P53 accumulation was present in the majority of cells Nuclear INI1-positivity was retained (Appendix A). The tumor was negative for glial fibrillary acidic protein (GFAP) and isocitrate dehydrogenase (NADP(+)) 1 (IDH1) R132H (Appendix A). The Ki67 proliferation index was up to about 50%. Interestingly, the *C11orf95-RELA* fusion could not be detected, and the tumor cells did not show nuclear accumulation of p65RelA protein.

### 2.3. Methylation Analysis Reveals a not Classifiable Tumor Entity

Due to the unexpected loss of the *C11orf95-RELA* fusion gene, we analyzed the primary tumor and the first metastatic relapse by 850k DNA methylation bead array analysis and the brain tumor classification tool recently described by Capper et al. (classifier version v11b4) [13]. The DNA methylation signatures of the primary tumor (no. 176), the metastasis (no. 225), and of the primary tumor cells isolated from the metastasis and grown in vitro (no. 225 ZL) did not show similarities with any known brain tumor DNA methylation class defined in this classifier version (Table 1), and thus were not classifiable by this method. A principal component analysis of genes conducted with the R package “RnBeads” indicated that the three samples cluster together, but not with *C11orf95-RELA* fusion gene positive ependymoma samples (Figure 2). These results argue for intermethodological discrepancies in the primary tumor, as RT-PCR and sequencing detected a *C11orf95-RELA* fusion gene, although 850k DNA methylation analysis did not show an association with the DNA methylation class of the *C11orf95-RELA* fusion gene positive ependymoma.

The copy number profile derived from the DNA methylation array data revealed a gain of chromosome 2 and chromosome 11q in the primary tumor (Table 1 and Appendix A). The first metastatic relapse further showed gains of chromosomes 1p, partially of 7q and 21q, and losses of chromosomes 17p and 19p. The copy number profiles of primary tumor cells showed an additional loss of chromosome 18q.

### 2.4. Whole-Transcriptome Sequencing Reveals the Activation of Several Embryonic Pathways

To identify therapeutic targets, whole-transcriptome sequencing (RNAseq) was performed using fresh frozen tissue from the first metastatic relapse (no. 225) and normal tissue derived from normal parietal brain (no. 111). To identify only strongly deregulated pathways, we first calculated the ratio of the expression between the tumor tissues and the normal tissue, and selected the genes with a fold change >10. This data set was functionally analyzed using the ‘core analysis’ of IPA. The activated pathways are shown in Table 2. The Notch and basal cell carcinoma (BCC) pathways had the lowest *p* value (*p* = 0.00024 and *p* = 0.0028, respectively). The BCC pathway is characterized by a cross-talk between the sonic hedgehog (SHH) and the wingless and integrated-1 (WNT) signaling [14]. Deregulated genes of the Notch and BBC pathways are listed in Appendix A and Appendix A, respectively. Other pathways activated in the relapse included the G12 subfamily (Gα12/13)-mediated signaling pathway [15] >(*p* = 0.0074). Since the first diagnosis of the tumor was of an ependymoma and IGF has been recently identified as relevant target in this entity [16], we also searched the transcriptome data for the expression of components of the IGF pathway. We observed a very strong expression of *IGF2*, but not of *IGF1* (Appendix A). In line with the results of the reference pathology, we were not able to detect a *C11orf95-RELA* fusion in the RNAseq data. However, we detected other fusions (Table 3), involving *nuclear receptor coactivator 1 (NCOA1)* and *GRB interacting GYF Protein 2* (*GIGYF2*) (both on chr.2) and *NCOA1* (chr.2) and *C11orf95* (chr.11). Fusions between *C11orf95* and *NCOA1*, a steroid receptor, have been described in a *C11orf95-RELA* negative supratentorial anaplastic ependymoma [17], but their biological significance is unknown so far. Two fusions contained intronic sequences, and are probably not functionally relevant. One fusion contained exon 8 of *GIGYF2*, which is a gene that is involved in the regulation of the IGF signaling [18]. Whether the disruption of the *GIGYF2* locus has an effect on the activation of the IGF signaling in this patient remains to be elucidated.

We further validated the RNAseq results by qRT-PCR using *GLI family zinc finger 2* (*GLI2*), *AXIN2*, and *hes family bHLH transcription factor 4* (*HES4*) as surrogate markers for the activation of the SHH, WNT, and Notch pathways, respectively. A strong upregulation of *GLI2*, *AXIN2*, and *HES4* was detected in the relapse compared to the two normal brain regions (Figure 3A–C). High expression of *IGF2* was also detectable in the relapse material by qRT-PCR (Figure 3D). In conclusion, the transcriptome analysis indicated a co-activation of several pathways known to play an important role in the tumor progression and embryogenesis.

### 2.5. Detection of a NOTCH 1 Germline Mutation

Due to the activation of the Notch signaling pathway, we searched for mutations in related genes in the RNAseq data, and detected a mutation in exon 3 of *notch receptor 1* (*NOTCH1*) (NM_017617.3:c.689G > A). Validation by Sanger sequencing indicated that the mutation was present not only in the tumor, but also in the lymphocytes of the patient (Figure 4A). The mutation is localized at the N terminus of NOTCH1 (NP_060087.3:*p*.Gly230Glu) in the fourth EGF-like domain according to cBioPortal [19] (Figure 4B), and is predicted to be damaging, according to polyphene [20]. A somatic mutation at this position has been described previously in a basal cell carcinoma patient [21]. In the DNA of the tumor, but not of the blood, a *TP53* homozygous mutation was also detected (NM_000546.5:c.742C > T, rs121912651) (Figure 4C). This mutation is localized in the DNA binding domain (NP_000537.3:p.Arg248Trp), and is predicted to be pathogenic according to ClinVar [22]. No mutations in *ALK* were detected by Sanger sequencing with primers covering the kinase domain [23].

### 2.6. The Primary Tumor Cells are Sensitive to Inhibition with ATO and Ceritinib

The transcriptome analysis revealed the SHH, WNT, Notch, and IGF pathways as putative targets for a personalized therapy protocol. We selected the pathways to treat based on the biological relevance of the pathways, safety concerns, and the availability of FDA-released drugs. To date, there is no approved Notch and WNT targeted therapy in the clinic, and the development of life-threatening toxicities associated with the targeting of these pathways still remain a concern [24,25]. The SHH pathway can be targeted using vismodegib, which is an inhibitor of smoothened (SMO) that has been approved for the treatment of the BCC [26], and ATO, an inhibitor of GLI2 [27,28]. The administration of ATO is well established in the treatment schedule of pediatric acute promyelocytic leukemia (APL) [29]. No inhibitors for IGF1R have been released so far, but it is known from clinical studies that the inhibition of the IGF pathway has an acceptable safety profile [30]. Moreover, ceritinib, which is approved for anaplastic lymphoma kinase (ALK) positive lung cancer, can also inhibit IGF1R and INSR [9]. We previously showed that ceritinib can inhibit the IGF receptor IGF1R in a particular subtype of pediatric brain tumors at a concentration that is achievable in vivo [11]. To evaluate if the targeting of the SHH and the IGF1R pathway could affect the growth of the tumor and to prioritize related drugs, we tested the effect of relevant drugs on the tumor cells (225ZL) isolated from the first relapse of the patient. We first analyzed to what extent the 225ZL cells constituted the tumor of origin. By DNA methylation analysis, the primary tumor cells were not assigned to any known brain tumor DNA methylation profile, but clustered together with the tumor of origin (Table 1 and Figure 2). Chromosomal aberrations were similar to the first metastasis with an additive loss of chromosome 18q (Table 1 and Appendix A). The Notch, SHH, and WNT pathways were active, as shown by qRT-PCR analysis of the target genes (Figure 3A–C). *IGF2* was highly expressed (Figure 3D). Then, we treated the 225ZL cells with ATO, vismodegib, and linsitinib, a specific but not FDA-approved inhibitor of IGF1R and INSR [31], and ceritinib. While vismodegib at a concentration of 10 µM reduced the cell proliferation to about 85% of the control, ATO, linsitinib, and ceritinib (all at the concentration of 1 µM) reduced the cell proliferation to about 16%, 20%, and 30% of the control respectively, after 15 days (Figure 5A). Notably, we selected the 1-µM concentration of ceritinib and linsitinib, because this concentration can be achieved in the plasma of patients [8,32], and is therefore clinically relevant. To identify the molecular target of ceritinib, we analyzed the expression of IGF1R and INSR by Western blot (Figure 5B). Only INSR was expressed by the 225ZL cells. Stimulation with IGF2 induced INSR phosphorylation in starved cells (Figure 5B, lane 2). The phosphorylation could be blocked using ceritinib (Figure 5B, lane 3). These data indicate that GLI2 and INSR are relevant targets to be addressed in a personalized therapy protocol, and that ceritinib can be used to target the IGF pathway.

### 2.7. Ceritinib Induces Tumor Regression in Vivo

Following the second relapse and in the absence of established treatment options, we switched to a personalized therapy protocol based on our molecular analysis of the tumor and our in vitro model (Figure 6). We decided to start with the inhibition of the SHH pathway, because the involvement of the SHH pathway in the biology of pediatric brain tumors is well established [33]. We did not use vismodegib to target the SHH pathway, because in our in vitro model, vismodegib reduced the cell proliferation to about 85% of the control, while the effect on the proliferation of ATO was much more impressive (about 16% of the control). Moreover, SMO inhibition causes permanent defects in bone structure in young mice and young children, and its use has to be carefully considered with respect to long-term toxicities [34,35]. As a chemotherapy backbone, we incorporated systemic VA (vincristine and actinomycin-D) and VAd (adriamycin) cycles and intraventricular (via Ommaya) cytarabine and etoposide-Gry. Arsenic trioxide synergizes with vincristine, doxorubicin, and etoposide [36,37]. Actinomycin-D was included to re-establish the tumor-suppressive function of TP53 [38]. The intraventricular therapy was based on the MEMMAT protocol (NCT01356290), a Phase II study of metronomic and targeted anti-angiogenesis therapy for children with recurrent/progressive medulloblastoma, that has also shown activity in different pediatric brain tumors [39].

ATO was administered to coincide with the chemotherapy backbone as an intravenous (iv) treatment for 1 week, and was subsequently switched to an oral ATO formulation [40] for a period of approximately 10 weeks in order to forgo hospitalization of the patient. Recently, we have experienced the clinical use of oral ATO in a pediatric brain tumor patient with BCOR alteration [28]. To reach therapeutically efficacious concentrations, we have chosen a higher dose than recommended for the treatment of pediatric patients with APL (0.2 mg/kg BW daily versus 0.15 mg/kg BW). ATO was initially well tolerated intravenously and orally. At week 9, the patient complained about violent leg pain in temporal association to the ATO administration, which was therefore interrupted. At the end of the treatment (67 days), the concentration of ATO was of 3.5 µg/L in the cerebrospinal fluid (CSF) and 30.5 µg/L in the plasma. This concentration was far below the expected concentration based on data of patients with APL and our own data on HGNET-BCOR [28,41], suggesting an incompliance of the patient due to the side effects, which was confirmed by the mother at a future date.

We switched to ceritinib for approximately 3.5 months. Due to the high malignancy of the disease, we initially decided to maintain the backbone of therapy in parallel to ceritinib for the first four weeks. Furthermore, it was unknown if the efficacy of the ceritinib monotherapy, which was successfully used to target ALK, was equivalent in the context of off-targeting IGF inhibition. The combination of IGF inhibition and chemotherapy is supported by clinical data showing that the inhibition of IGF signaling can enhance the effects of chemotherapy [42]. The combined therapy was maintained until the detection of the new cerebellar lesion (blue star in Figure 6). At this point, we decided to apply ceritinib alone. The rationale was on one hand, that no further benefit was observed, and on the other hand, that so far, no clinically proven interaction of ceritinib with chemotherapy was known. Ceritinib has been shown to be effective as monotherapy in brain metastasis of lung cancer (with a median time to intracranial response of 6.1 weeks [43]). Additionally, we intended to improve the quality of life of the patient, avoiding further hospitalization. Ceritinib was applied as opened capsules, dissolving pH-dependent (pH 4.2) in commercially available cherry–banana juice according to the staff recommendation of the Hospital for Sick Kids, Toronto, Canada, to facilitate the intake of oral chemotherapy in children. Due to the well-known side effects of nausea and vomiting caused by ceritinib, the patient merely received 350 mg/m² body surface area in preference to the recommended without-food dosage of 510 mg/m² body surface area in other pediatric studies (NCT01742286). Beside nausea, no further toxicity related to the ceritinib treatment was observed, including cardiac, liver, lung, or gastrointestinal symptoms, which were investigated weekly by ECG and blood works. After 30 days of ceritinib monotherapy, a response to the treatment was confirmed by MRI (Figure 7). The larger right temporal lesion developed diffuse hemorrhagic transformation under therapy with ceritinib (Figure 7, A/B versus D/E) and was removed surgically 10 days later. Histopathology demonstrated near complete necrosis as well as bleeding with only scattered areas of viable cells. Bleeding and necrosis have been previously described in association with response to IGF1R inhibitors in relapsed malignant astrocytoma and squamous non-small cell lung carcinoma [44,45]. It remains to be clarified if an off-target inhibition by ceritinib is comparable to an inhibition with specific IGF inhibitors in term of adverse events such as hemorrhage. Moreover, a smaller cerebellar lesion has reduced contrast uptake within one month under therapy with ceritinib (Figure 7, C versus F, magnified images). Following the detection of the response by MRI, the patient refused the periodic intake of ceritinib, and post-surgical MRI revealed a third relapse 14 days after the detection of the response. The patient died 27 months after initial diagnosis and 3.5 months after the third relapse.

### 2.8. Ceritinib Penetrates the Human Brain

To assess the concentration of ceritinib, we developed and validated an UPLC/MS/MS-based assay. The patient’s plasma concentrations reached a peak of 1.70 µM after 46 days (the concentration described in the literature is ~1 µM [8]). This peak plasma level of ceritinib was observed at time point d 12 before MRI response was detected. Following the detection of the response, the patient refused the periodic intake of the medication, and the plasma level decreased until 0.9 µM. The post-surgical third relapse followed approximately 2 weeks of markedly diminished ceritinib levels. Interestingly, no ceritinib (< 2.5 ng/mL) was detectable in the CSF samples that were collected in parallel to the plasma samples. However, after the surgical removal of the sanguineous necrotic area, we quantified a ceritinib concentration of 1516 ng/g (2.72 µM) in the isolated tumor areal, while the concentration of ceritinib in the plasma was 0.9 µM one day before the surgery. These results indicated that ceritinib was able to cross the blood–brain barrier (BBB) and reached effective concentrations in the tumor.

## 3. Discussion

Numerous challenges remain in the development of molecularly targeted therapies, including the identification of meaningful targets, delivery of agents at the target site, and the determination of response to these agents. This is particular true for pediatric patients in whom underlying germline mutations can contribute to unusual histologies. The percentage of brain cancer in children and adolescents attributable to an underlying genetic syndrome or inherited susceptibility is estimated at 8.6% [46]. Germline mutations in *NOTCH1* cause aortic valve disease [47] and have been found to cause Adams-Oliver syndrome [48]. All these alterations are associated with a downregulation of the Notch signaling, while in our patient, the Notch signaling was upregulated. To our knowledge, *NOTCH1* germline mutations have not been associated with cancer so far. However, somatic *NOTCH1* mutations have been described in hematopoietic and solid tumors [49,50,51]. Notably, the Notch pathway is implicated in ependymoma oncogenesis [52], and *NOTCH1* mutations have been described to present in 8.3% of pediatric ependymomas samples in one study [53]. The mutation described here is localized in the EGF-like domain of NOTCH1, which protects NOTCH1 from proteolytic cleavage inhibiting uncontrolled constitutive activation, and governs ligand-induced homodimerization [54]. It remains to be clarified if this mutation was the driver element for the activation of the Notch pathway in the tumor of the patient described in this work. As an alternative, the Notch pathway could also be indirectly activated via the WNT pathway [55]. The germline *NOTCH1* mutation was accompanied by a somatic *TP53*-inactivating mutation. Notably, *Notch1* expression results in SHH medulloblastoma formation in p53-deficient mice [56], suggesting a synergistic effect of both mutations in the malignancy of the disease, and a possible cross-talk between the Notch and SHH pathways in the patient’s tumor.

Developmental pathways, such as those regulated by IGF, Hedgehog, WNT, and Notch, play key roles in the controls of cell fate decisions during development. A cross-talk among these signaling pathways is typical for cancer stem cells, and targeting embryonic signaling pathways is a current challenge in cancer therapy [57]. In brain tumors, IGF1R is frequently activated in medulloblastoma [3], and INSR and IGF1R are frequently activated in glioblastoma [5]. To date, neither monoclonal antibodies nor small molecule tyrosine kinase inhibitors directed against the IGF pathway have been approved. Ceritinib inhibits kinases such as ROS proto-oncogene 1 (ROS1), INSR, and IGF1R, although it is most active against ALK [9]. Lung cancer patients with brain metastases also respond to the treatment with ceritinib, suggesting that this drug is effective across the BBB [58]. The concentration achieved in the mouse brain after oral administration is low, with a brain-to-plasma ratio below 0.3 [59], but the blood-to-brain exposure ratio of ceritinib in humans is yet to be determined. We were not able to detect ceritinib in the CSF; however, we detected a high concentration in the tumor, suggesting that ceritinib crosses the BBB and is actively transported to the tumor. Ceritinib is a substrate of the efflux transport protein P-gp, which is in discussion to be an effective resistance mechanism against ceritinib [60], and which is located in the BBB [61]. From this point of view, low ceritinib brain concentration appears to be a reasonable consequence. However, the results in this patient reveal about doubled ceritinib tumor concentration compared to plasma, although ceritinib was not detectable in the CSF. Leakiness of the BBB has been described, particularly in patients with a WNT-activated brain tumor, and this could explain the effective penetration of ceritinib in the brain in this patient [62]. The accumulation in the tumor rather than in the CSF may be explained by a parallel efficient uptake transport, although it is reported, that ceritinib is not a substrate of breast cancer resistance protein (BCRP), organic cation transporter 1 (OCT1), organic anion transporter 2 (OAT2), or organic anion transporting polypeptide 1 (OATP1) [63]. Therefore, the specific and irreversible binding of ceritinib to so-far unknown intracellular binding sites in the tumor seems to be the most reasonable explanation.

After a first response to ceritinib, the tumor progressed. Although we cannot rule out whether the reduced intake of ceritinib by the patient after detection of the response may have contributed to the progress, it is already known that several mechanisms are associated with the resistance to IGF pathway inhibition. These mechanisms include the compensatory actions of other growth factor receptors [30]. In the tumor of the patient described in this work, other embryonic pathways were upregulated, and it is possible that the tumor cells switched to other signaling to overcome INSR blockade. During the course of disease, the tumor displayed rapid clonal evolution, so that a selection of resistant cell clones is likely. The identification of distinctive mechanisms of resistance in this particular patient is ongoing. In conclusion, here we present the targeting of the IGF receptor INSR using ceritinib in a patient with an incurable neuroepithelial brain tumor. Due to the limitations of our work, which included only one patient with an unknown histology; a systematic investigation of ceritinib for the treatment of IGF-driven tumors in the framework of a clinical study is imperative.

## 4. Materials and Methods

### 4.1. Tissue Samples and Cells

Fresh tumor material from the first relapse (no. 225) was obtained during standard surgery. The RNA of normal brain tissues (adult frontal lobe no. 110 and adult parietal lobe no. 111) was sourced from commercial vendors (Biocat, Heidelberg, Germany). For the isolation of a primary cell culture (225ZL), the fresh tumor sample (no. 225) was disrupted with the GentleMACS Dissociator (Miltenyi Biotec GmbH, Bergisch-Gladbach, Germany), and the cells were dissociated with 0.25% trypsin. The cells were cultured in DMEM medium containing 10% human serum, 1% l-glutamine, and 1% penicillin-streptomycin (all Sigma-Aldrich, Taufkirchen, Germany). Cell at passage 5 were used for further experiments. This study has been performed in accordance with the ethical standards laid down in the 1964 Declaration of Helsinki and its later amendments. The parents gave their informed consent prior to their inclusion in the study. Formal approval of the local ethics committee for this study was not required, as this was a single case investigation. For details about the ethics, see the supplemental material and methods.

### 4.2. Nucleic Acid Extraction

Tumor samples were analyzed by a pathologist, and regions containing vital tumors were isolated for further processing. DNA from blood and tissues was extracted using a Gentra Puregene Blood Kit (Qiagen, Hilden, Germany). RNA isolation was conducted using the RNeasy Lipid Tissue Mini Kit (Qiagen). RNA was converted to cDNA by using PrimeScript RT Reagent Kit with gDNA Eraser (Takara Bio Europe, Saint-Germain-en-Laye, France). Quality control was performed using a Bioanalyzer2100 (Agilent Technologies, Waldbronn, Germany). Only RNAs with a RIN value > 7 were used for RNA sequencing analysis and qRT-PCR.

### 4.3. DNA Sequencing

PCR products were sequenced using an ABI Prism 3100 Genetic Analyzer and the BigDye v3 Terminator Kit (Thermo Fisher, Dreieich, Germany). The sequences were compared to the reference sequence using the Sequencher program (Gene Codes, Corp., Ann Arbor, MI, USA). The following primers were used: *NOTCH1*: 5′- GATGTCAACGAGTGTGGCCA and 5′- AGTTCCGGGAAACTCCAGAGA; *TP53*: 5′- GCTTGCCACAGGTCTCCC and 5′- GAGGCAAGCAGAGGCTGG.

### 4.4. qRT-PCR

qRT-PCR was performed using the Light Cycler 480 II detection system and software (Applied Biosystems, Darmstadt, Germany) with a KAPA SYBR FAST Light Cycler 480 Kit (PeqLab, Erlangen, Germany). The following primers were used: *AXIN2*: 5′-GCTCAGAGCTTGACCCTGG and 5′- TCATACATCGGGAGCACCGT; *HPRT1*: 5′-TGACACTGGCAAAACAATGCA and 5′-GGTCCTTTTCACCAGCAAGCT. *GLI2*: 5′-TCCACACACGCGGAACACCA and 5′- CAGCTGGCTCAGCATGGTCA, *HES4*: 5′-GTGCAGGTGACGGCCGC and 5′- CGGCCAGGAAGCGGTTCA.

### 4.5. RNA Sequencing

Libraries were constructed using the TruSeq mRNA stranded protocol (Illumina, San Diego, CA, USA) using 2 μg of total RNA. Pair-end sequencing with a read length of 150 bps was performed on a NextSeq500 Instrument (Illumina, San Diego, CA, USA). A total of 50 million reads per library were produced. Read mapping was performed using the TopHat2 v2.0.7 aligner [64] and the Homo sapiens UCSC hg19 reference genome (RefSeq gene annotations). The FPKM values of reference genes and transcripts were estimated by Cufflinks 2.0.5 [65]. TPM values of genes and transcripts were calculated according to Lior Pachter (https://arxiv.org/abs/1104.3889). Ingenuity pathway analysis (IPA; Ingenuity Systems/Qiagen, Redwood City, CA, USA) was used to map lists of significant genes to biological pathways. To identify mutations, data were processed using BWA Enrichment v1.0 for the generation of BAM files and GATK for variant calling. Translocation was identified with TopHat-Fusions [66]. Analysis of variants was performed with the VariantStudio software (Illumina, San Diego, CA, USA).

### 4.6. Methylation Analysis and Copy Number Variation Profiles

DNA methylation analyses were performed using an EPIC 850k DNA methylome bead array (Illumina, San Diego, CA, USA). We used standard protocols for tissue and DNA processing. Hybridization and processing of the chip was performed as indicated by the manufacturer. Data were preprocessed using Illumina Genome Studio; further analysis was performed after uploading raw data (idat files) onto the platform MolecularNeuroPathology.org 2018 version 3.1.2, which provides brain tumor classifier results (version v11b4) and copy number variation profiles. Furthermore, we compared the DNA methylation profiles of the unclassified tumor specimens with tumor tissue deriving from common childhood brain tumor entities by applying principal component analysis using the R package “RnBeads” [67].

### 4.7. Cellular Proliferation Assays

Linsitinib, vismodegib, and ceritinib (Selleck Chemicals, Houston, TX, USA) were dissolved in DMSO (Sigma-Aldrich, Taufkirchen, Germany) to a final concentration of 10 mM. Arsenic trioxide (ATO) was prepared as previously described [68]. 225ZL cells were plated in triplicates at a density of 5000 cells/well in a 96-well plate, and incubated with the different drugs. Viable cells were quantified using the cell proliferation reagent WST-1 (Roche, Mannheim, Germany).

### 4.8. Phosphorylation Assay

225ZL cells were grown in charcoal-stripped (Sigma-Aldrich, Taufkirchen, Germany) DMEM supplemented with 10% human serum, 2 mM of L-Glutamine, 1 mM of sodium pyruvate, and penicillin–streptomycin for 1 day. The cells were serum-starved for 24 h, and then treated with 1 µM of ceritinib for two hours, followed by incubation with 20 ng/mL of IGF2 for 15 min. Cell lysates were loaded on SDS-PAGE gels, followed by blotting to a polyvinylidene difluoride membrane (BioRad Laboratories, Inc., Hercules, CA, USA). The following antibodies were obtained from Cell Signaling Technology (Cambridge, UK): GAPDH (14C10) (cat # 2118, 1:1000 dilution), IGF1 receptor β (D23H3) XP^®^ (cat # 9750, 1:1000 dilution) and phospho-IGF1 receptor β (Tyr1131)/Insulin Receptor β (Tyr1146) (cat # 3021, 1:1000 dilution). Detection was done by a SuperSignal™ West Dura Extended Duration Substrate (Thermo Fisher Scientific, Waltham, MA, USA), and the imaging was performed on Fusion Pulse TS (Vilber Lourmat, Eberhardzell, Germany).

### 4.9. Targeted Therapy Protocol

The patient was enrolled for genomic profiling following the parent’s informed consent. Tumor and blood samples were then acquired, processed, molecularly profiled, and analyzed computationally. Molecular results were reviewed in the in-house pediatric tumor board. No early-phase clinical trial was available for this patient. The parents have been informed of all the disease and treatment-related relevant molecular results. Targeted therapy was initiated following parent’s consent regarding compassionate and off-label use. For protocol details, see the results section.

### 4.10. Arsenic Concentration

The total arsenic concentration in CSF and plasma was analyzed as described in [28].

### 4.11. Ceritinib Quantification

Ceritinib concentration in plasma, CSF, and tumor tissue was quantified using UPLC-MS/MS- and liquid/liquid extraction techniques. For quantification in the tumor tissue, regions containing vital tumors were isolated from the surrounding necrotic regions by a pathologist. The plasma, CSF, and tissue samples were spiked with internal standards d7-ceritinib and boric buffer pH 9 (50 µL). Subsequently tert-butylmethylether was added, shaken for 10 min, and centrifuged (10 min, 3000 g). The supernatants were evaporated to dryness in a stream of nitrogen at 40 °C, reconstituted by adding LC eluent (100 µL), and injected (20 µL) into the UPLC-MS/MS system, which consisted of an Acquity sample manager, an Acquity solvent manager, and a TQD triple stage quadrupole mass spectrometer (Waters GmbH, Eschborn, Germany). For chromatographic separation, an Acquity CSH C18 (1.7 µm; 2.1 × 50 mm) column (Waters GmbH, Eschborn, Germany) at 40 °C was used. The eluent consisted of water, including 0.01% formic and 5% acetonitrile (A) and acetonitrile including 0.01% formic acid (B). For separation a gradient program at 0.5 mL/min was applied. From 0 to 0.5 min, 95% A/5% B was used. From 0.5 to 2.5 min, the ratio was linearly changed to 5% A/95% B and held for 3.5 min. The eluent was introduced directly into the electrospray ion source of the tandem mass spectrometer (MS/MS). The MS/MS transitions monitored in the positive ion mode were m/z 558.1 → m/z 84.0 at 38 V for ceritinib and m/z 565.1 → m/z 84.0 at 38 V for d7-ceritinib. The assay was validated according to common FDA and European Medicines Agency (EMA) validation guidelines on bioanalytical method validation. The lower limit of quantification of ceritinib was 2.5 ng/mL. The calibrated range was 2.5–1000 ng/mL (4.5–1790 nM) with correlation coefficients > 0.995. The overall accuracy varied between –2.3% and +5.2%, and an overall precision ranging from 2.3% to 12.4%.

## Figures and Tables

**Figure 1 ijms-20-04267-f001:**
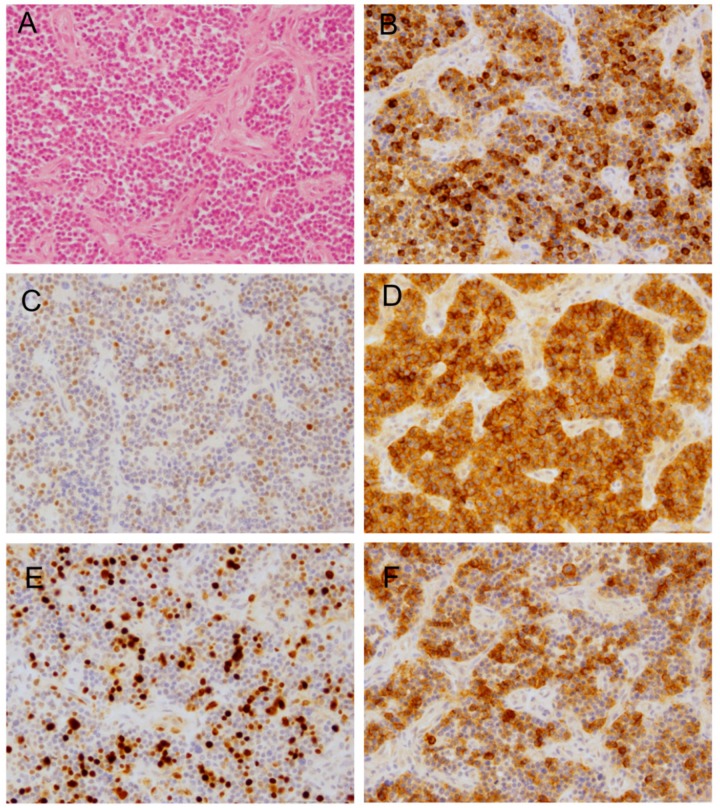
Histopathological features of the primary tumor. (**A**) HE staining showing small, round, blue tumor cells. (**B**) Epithelial antigens (EMA). (**C**) NeuN. (**D**) CD56. (**E**) Ki67. (**F**) L1Cam. Original magnification 200×.

**Figure 2 ijms-20-04267-f002:**
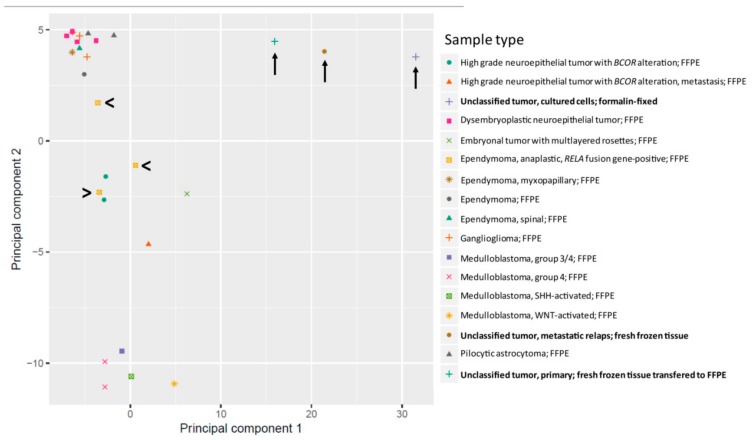
Principal Component Analysis. Principal component analysis by genes based on 850 k DNA methylation analysis for different tumor entities commonly found in childhood. Samples of the index patient do not cluster together with ependymoma, *RELA* fusion-positive tumors (arrow heads), but rather form their own cluster (arrows). The kind of material used for the analysis is indicated (fresh frozen or formalin-fixed, paraffin-embedded (FFPE)).

**Figure 3 ijms-20-04267-f003:**
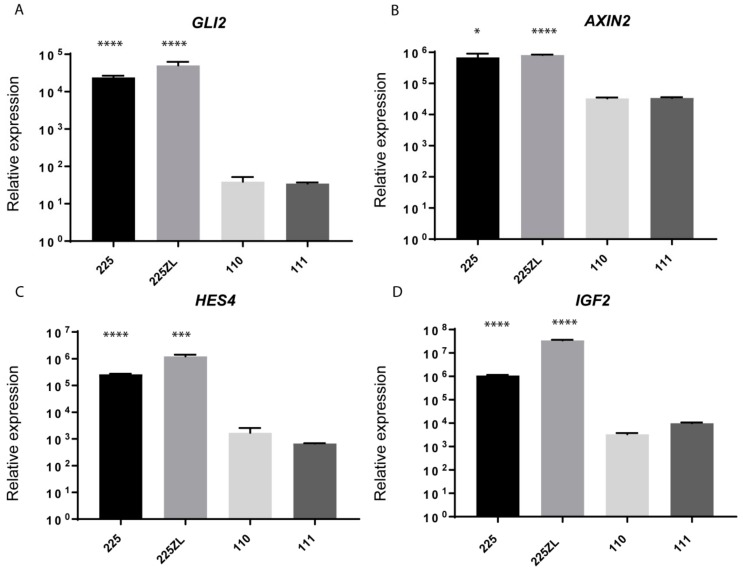
Embryonal pathways are activated in the tumor relapse. *GLI family zinc finger 2* (*GLI2*) (**A**), *AXIN2* (**B**), *hes family bHLH transcription factor 4* (*HES4*) (**C**) and *insulin-like growth factor 2* (*IGF2*) (**D**) expression was analyzed by qRT-PCR in the relapse (no 225), the primary tumor cells isolated from the relapse (no 225ZL), and two normal brain regions (no 110, 111). After normalization to the housekeeping gene *hypoxanthine phosphoribosyltransferase 1* (*HPRT1*), the relative quantification value was expressed as 2^-ΔΔ*C*t^. The qPCR experiments were carried out in biological triplicates. Data are represented as the mean ± standard deviation (SD). Statistical analyses were performed using *t*-tests compared with sample 111 (* *p* < 0.01, *** *p* < 0.001, **** *p* < 0.0001).

**Figure 4 ijms-20-04267-f004:**
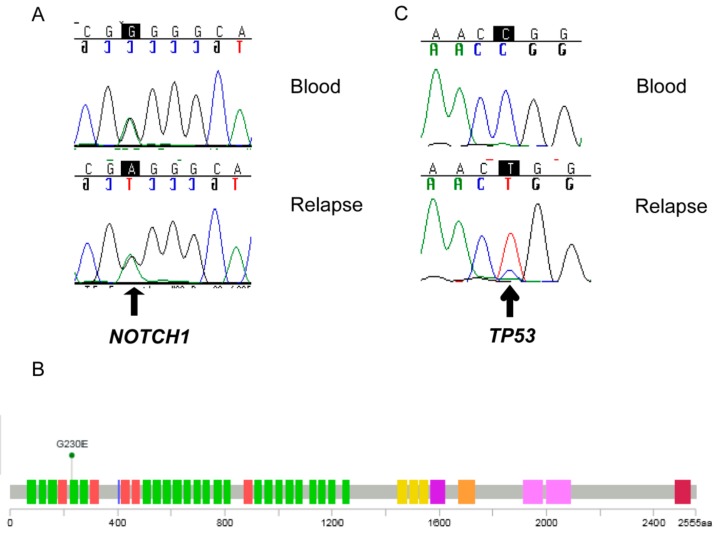
Detection of a *notch receptor 1* (*NOTCH1*) germline mutation and a tumor protein P53 (*TP53*) somatic mutation. Sanger sequencing of the genomic DNA extracted from the blood and the relapse (no 225) was done with primers specific for the *NOTCH1* (**A**) and the *TP53* (**C**) mutation identified by RNA-seq. The arrow indicates the (NM_017617.3:c.689G > A) mutation rs121912651. The arrowhead indicates the rs121912651. (**B**) Structure of NOTCH1 as reported in cBioportal. The position of the Gly230Glu mutation is indicated. Green: EGF-like domains; Red: Calcium-binding EGF domains; Yellow: LNR Domain; Purple: NOD (NOTCH protein domain); Orange: NODP (NOTCH protein domain); Rose: Ankyrin repeats; Dark red: unknown function.

**Figure 5 ijms-20-04267-f005:**
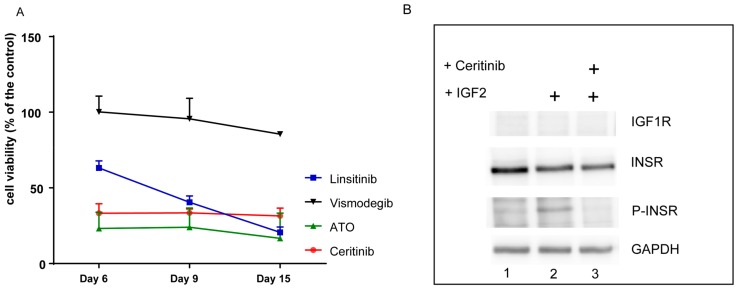
The primary tumor cells are sensitive to arsenic trioxide (ATO) and ceritinib. (**A**) 225ZL cells were grown for 15 days in the presence of ATO, vismodegib, linsitinib, or ceritinib. As control, the vehicle alone was used. The percentage of the growth compared to the control is shown. The proliferation experiments were carried out in biological duplicates. Data are represented as the mean ± standard deviation (SD). (**B**) 225ZL cells were stimulated after starvation with IGF2 in the presence or absence of 1 µM of ceritinib. The expression of IGF1R, insulin receptor (INSR), and the phosphorylated form of INSR (P-INSR) was analyzed by Western blot with specific antibodies. Glyceraldehyde-3-phosphate dehydrogenase (GAPDH) was used as loading control. A representative experiment of two independent experiments is shown.

**Figure 6 ijms-20-04267-f006:**
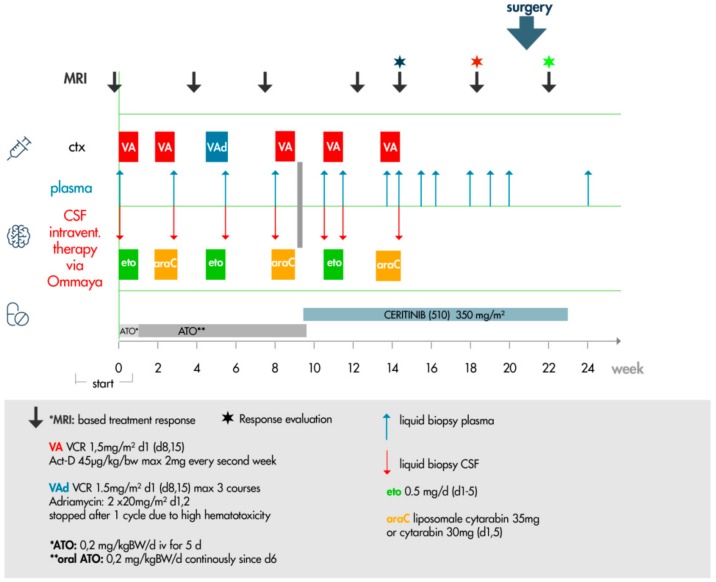
Personalized therapy protocol. Chemotherapy incorporated systemic (ctx) VA (vincristine and actinomycin-D, red) and VAd (adriamycin, blue) blocks, and intraventricular (via Ommaya) cytarabine (araC, yellow) and etoposide-Gry (eto, green). ATO was administered to coincide with the chemotherapy backbone as an intravenous (iv) treatment for 1 week (bright grey), and was subsequently switched to an oral ATO formulation (dark grey). The duration of ATO treatment lasted 68 days (9.7 weeks) in total. Ceritinib was given for 93 days. A metastatic lesion in the cerebellum was detected by MRI after 30 days of ceritinib treatment in combination with VA and intraventricular therapy (blue star). After 30 days of ceritinib monotherapy following the detection of the metastatic relapse, a response to the treatment was confirmed by MRI (red star). Surgical removal of the sanguineous necrotic area was performed (big blue arrow). A third relapse was detected by MRI (green star) two weeks after the detection of the response. The patient died 27 months after initial diagnosis and 3.5 months after the third relapse.

**Figure 7 ijms-20-04267-f007:**
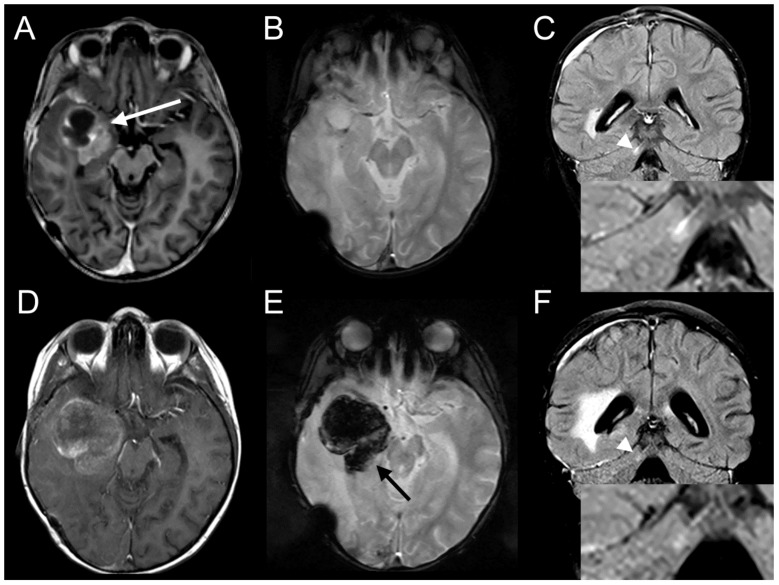
The target lesion is sensitive to the targeted therapy with ceritinib. MRI before therapy with ceritinib (**A**–**C**) showing a right sided solid mesiotemporal tumor (**A**, white arrow) with a cystic component, as well as a contrast media enriching metastatic lesion within the cerebellar folia (**C**, FLAIR post-contrast media, white arrow head). After 30 days of ceritinib (**D**–**F**), the metastatic lesion was regredient (**F**, FLAIR post-contrast media, white arrow head), and hemorrhagic transformation (**E**, T2*, black arrow) of the tumor was detected.

**Table 1 ijms-20-04267-t001:** Results of the methylation classifier and summary of the chromosomal aberrations.

Sample Type	Sample ID	Methylation Classifier Result	Chromosomal Aberrations
Primary tumor	176	unclassified	Gain on chromosome 2 and 11q
First metastatic relapse	225	unclassified	Tendency to gain on chromosomes 1p, 2, 7q partially, 11q, 21q; loss of chromosomes 17p, 19p
in vitro culture of tumor cells	225 ZL	unclassified	Gain on chromosomes 1p, 2, 7q partially, 11q, 21q; loss of chromosomes 17p, 18q, 19p

**Table 2 ijms-20-04267-t002:** Pathways activated in the first metastatic relapse. The -log of *p*-value (calculated by Fisher’s exact test right-tailed) is indicated. Only pathways with a -log of *p*-value of more than 2 are shown.

Ingenuity Canonical Pathways	-Log (*p*-value)
Notch Signaling	3.62
Basal Cell Carcinoma Signaling	2.55
Gα12/13 Signaling	2.13

**Table 3 ijms-20-04267-t003:** Fusion transcripts detected by RNA Seq.

Transcript 1	Chr	Position	Transcript 2	Chr	Position
*NCOA1* intron7	2	24914529–24916118	*GIGYF2* exon8	2	233626104–233626145
*NCOA1* intron12	2	24933980–24949455	*GIGYF2* intron8	2	233626146–233651857
*NCOA1* intron12	2	24933980–24949455	*C11orf95* intron3	11	63532726–63533276

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
