# Peer review of "Ceritinib-Induced Regression of an Insulin-Like Growth Factor-Driven Neuroepithelial Brain Tumor"

_ijms, 2019, doi:10.3390/ijms20174267_

Round 1

Reviewer 1 Report

Russo and colleagues provide an interesting case study on one pediatric (ependymoma) tumor including histological and molecular profiling of the tumor's subtype and development. They also report on the clinical response of the patient to an experimental treatment regime. In my opinion, the study is potentially publishable with editorial modifications:

The study conclusion paragraph (~from line 350) must include a clear statement of the study’s limitations (e.g., n=1; ‘own’ rather than ‘RELA-fusion-positive’ tumor classification). The current conclusion ‘ceritinib might be a beneficial substance for IGF-driven tumours' is not fully substantiated and should be’ toned down’. 

Results section (104): data not shown is not acceptable  - please show the data (GFAP-negative staining etc.) in the supplementary materials.

Properly introduce all acronyms and gene nomenclatures (e.g., GLI2, AXIN2, HES4).

Avoid cryptic axis labelling, e.g., Figure 5: I guess this shows the percentage of overall cell viability?

State the exact p-value (line 151).

Include statistical analysis into figures and legends wherever possible and clarify whether experimental repeats were of technical or biological nature (e.g., clarify ‘triplicates’ in the figure 3 caption).     

Additional proofreading is required. Examples for attention: ‘factor - driven (in title)’, ‘(DTK),69120 (affiliation section)’,  ‘350mg (272)’, 'tumor.3.Discussion (299)’.

As the study was not subject to local ethical approval, please clarify how the consent procedure matched current ethical standards and how patient anonymity was achieved (i.e., in terms of how potential clinical/academic boundary breaches were avoided).

Author Response

The study conclusion paragraph (~from line 350) must include a clear statement of the study’s limitations (e.g., n=1; ‘own’ rather than ‘RELA-fusion-positive’ tumor classification). The current conclusion ‘ceritinib might be a beneficial substance for IGF-driven tumours' is not fully substantiated and should be’ toned down’. 

As fairly suggested by the reviewer, we have included a statement on study limitation (from line 405):

In conclusion, here we present the targeting of the IGF receptor INSR using ceritinib in a patient with an incurable neuroepithelial brain tumor. Due to the limitations of our work, which included only one patient and an unknown histology, a systematic investigation of ceritinib for the treatment of IGF driven tumors in the framework of a clinical study is imperative”

Results section (104): data not shown is not acceptable  - please show the data (GFAP-negative staining etc.) in the supplementary materials.

We have added the staining of GFAP and IDH1 R132H in Supplemental Figure 1

Properly introduce all acronyms and gene nomenclatures (e.g., GLI2, AXIN2, HES4).

We have introduced all acronyms and gene nomenclatures according to HUGO

Avoid cryptic axis labelling, e.g., Figure 5: I guess this shows the percentage of overall cell viability?

We have changed the name of the y axis in Figure 5 in cell viability (% of the control)

State the exact p-value (line 151).

We have added the p value in the text: “The Notch and Basal Cell Carcinoma (BCC) pathway had the lowest p value (p = 0.00024 and p =0.0028 respectively)” (line 156), and “Other pathways activated in the relapse included the G12 subfamily (Gα12/13)-mediated signaling pathway  (p = 0.0074)” (line 159).

Include statistical analysis into figures and legends wherever possible and clarify whether experimental repeats were of technical or biological nature (e.g., clarify ‘triplicates’ in the figure 3 caption).     

We have included statistical and clarified whether experimental repeats were of technical or biological nature as follow:

Figure 3: The qPCR experiments were carried out in biological triplicates. Data are represented as the mean ± Standard deviation (SD). Statistical analyses were performed using t-tests compared with sample 111 (*p < 0.01, ***p<0.001, ****p < 0.0001).

Figure 5: The proliferation experiments were carried out in biological duplicates. Data are represented as the mean ± Standard deviation (SD).

Additional proofreading is required. Examples for attention: ‘factor - driven (in title)’, ‘(DTK),69120 (affiliation section)’,  ‘350mg (272)’, 'tumor.3.Discussion (299)’.

We thank the reviewer for the hint and we performed additional proofreading

As the study was not subject to local ethical approval, please clarify how the consent procedure matched current ethical standards and how patient anonymity was achieved (i.e., in terms of how potential clinical/academic boundary breaches were avoided).

The study delineates the administration of a pharmaceutical drug for an unapproved indication in an unapproved age group (off-label use). The purpose of off-label use is to benefit the individual patient and therapeutic decision-making is guided by the best available evidence and the importance of the benefit for the individual patient. In our patient, that suffered from a life-threatening disease, the off-label use of Ceritinib was performed as a individual therapeutic approach comparable to an compassionate use (except that Ceritinib is a drug approved by the EMA e.g. for the therapy in ALK-rearranged NSCLC in adults) lacking any other remaining evidence based treatment options. The European Medicines Agency (EMA)’s Guideline on Compassionate Use of Medicinal Products, Pursuant to Article 83 of Regulation (EC) No 726/2004, developed by the Committee for Medicinal Products for Human Use (CHMP) [1], states that compassionate use is performed primarily for therapeutic purposes of “patients with a chronically or seriously debilitating disease, or a life threatening disease, and who cannot be treated satisfactorily by an authorised medicinal product” in the European Union, of an individual patient. Thus, by its very nature, compassionate use represents a kind of treatment and not biomedical research [2], especially if not performed as a compassionate use series or program. Off-label and compassionate use regulations have already been introduced in Germany and do not require independent ethical review. The main focus of the German research ethics committee’s (REC) is to protect the rights, dignity and safety of biomedical research participants within clinical trials. Therefore, off-label and compassionate use concerning the treatment of an individual patient does not fall within the scope of interest of these committees.

In our case, the whole team and especially the physicians, who were responsible for the treatment, were regularly trained in Good Clinical Practice (GCP) standards and long-term experienced in conducting clinical trials. Furthermore, all treating physicians were qualified pediatric oncologists and members of the German pediatric society for hematology and oncology (GPOH) with raised awareness of the ethical aspects of compassionate use.

In order to protect the patient and to improve knowledge about medicine used in conditions other than those authorised, this patient was closely (weekly) observed, suspected adverse drug reactions (ADR) were reported in a timely fashion, and safety signals were identified and monitored.

The informed consent process comprised a multi-step-procedure concerning the enlightenment of the parents by the medical team in the presence of an innocent bystander and a non-medical-versed native speaker for the translation into english for the father. Following each session, the parents were proactively encouraged to obtain a second opinion. This approach was utilized twice.

We used informed consent forms that are standardized, especially designed for off-label use, approved for the application in our institution and commercially available (Thieme Compliance GmbH, Erlangen, Germany 2017 version 10/72017v1, order number DE613126).

Prior to the enlightening of the parents, the treatment was discussed with colleagues of the German relapse brain tumor study coordination office, HIT Rez, Essen, Germany. This statement, determining the favorable risk: benefit ratio, as well as the biological in vitro data, constituted the basis for an application for the cost takeover by the patients health care insurance.

German law includes some specific provisions on off-label use in the regulation on health insurance (Paragraph 275 social secure code, §275 SGB V [3]: therefore, a special medical advisory service of the German association of the statutory health insurance funds (MDK), consisting of independent experts in oncology, reviewed the application. Finally, their statement led to a positive vote for the cost takeover by the patients health care insurance. So, the decision for using ceritinib was made independently of any commercial sales interests of the drug manufacturer.

Concerning the anonymity of the patient, samples were stored in pseudonymization form. Pseudonymization adds an additional layer of protection for person-related data. It has been implemented as important security measure in many projects including the German National Cohort. Only authorized persons have access to the data.

We have included these aspects in supplemental material and methods

Literatur

1 https://www.ema.europa.eu/en/documents/regulatory-procedural-guideline/guideline-compassionate-use-medicinal-products-pursuant-article-83-regulation-ec-no-726/2004_en.pdf Accessed 27.7.2019

2 Borysowski et al. BMC Medicine (2017) 15:136

DOI 10.1186/s12916-017-0910-9.

3 https://www.gesetze-im-internet.de/sgb_5/__275.html accessed 27.07.2019

Reviewer 2 Report

The authors report the use of certinib to treat a 4-year-old patient diagnosed with metastatic progression after resection of a right parieto-occipital anaplastic ependymoma. In order to facilitate a personalized approach to treatment, the primary tumor tissue and cultured brain tumor cells were characterized in an epigenetic, genomic, and transcriptomic manner. Cultured tumor cells were used to screen responses to different agents that targeted the altered signalizing pathways found on molecular analysis of the tumor.

Personalized treatment of tumors is on the rise, and the authors did a thorough job of analyzing the patient's tumor tissue. However, I can some concerns about methodology and conclusions drawn that are listed below.

- The authors did compare the cultured tumor cells to the primary tumor tissue using DNA methylation and RT-PCR. However, the latter only looked at a few genes. Comparing the cultured cells and primary tissue via RNA-Seq would have provided a more complete demonstration of the similarity between the two samples. Showing this point is particularly important in using in vitro models to screen drug responses. Some details about the cultured cells appear to be missing. What passage number were the cultured cells? If later passages were used, is there data showing that the molecular profile of the cells do not drift over time?

- How did the authors decide on which altered signaling pathways to treat? For example, vismodegib appeared to be effective in the in vitro screen, but there is no mention of why this drug was not used for the patient.

- Was the backbone chemotherapy used concurrently with targeted therapy the same or different as after the first recurrence? If different, why was it thought that additional targeted therapy was needed at the same time? The use of so many different agents makes it more difficult to tell the effective of the target therapy. Along these lines, it is not clear why the background chemotherapy was abandoned in favor of ceretinib monotherapy.

- The imaging and text are not completely clear about which lesion was responsive to ceretinib monotherapy. If it is the cerebellar lesion that is thought to have responded, this response does not appear to be very significant. Is the FLAIR signal in the cerebellum entirely the tumor, or could it be edema associated with a very small lesion. Could it be that differences in the FLAIR signal are related to changes in edema instead of tumor? Also, the coronal slice is not quite the same pre- versus post-therapy, and I would be concerned that the slight difference in size of the cerebellar FLAIR signal could just be because of different locations are being looked at.

Author Response

The authors did compare the cultured tumor cells to the primary tumor tissue using DNA methylation and RT-PCR. However, the latter only looked at a few genes. Comparing the cultured cells and primary tissue via RNA-Seq would have provided a more complete demonstration of the similarity between the two samples. Showing this point is particularly important in using in vitro models to screen drug responses. Some details about the cultured cells appear to be missing. What passage number were the cultured cells? If later passages were used, is there data showing that the molecular profile of the cells do not drift over time?

We agree that RNA seq analysis would provide a more complete comparison between the primary cells and the tumor of origin. However, in this study we aimed to show that selected pathways identified in the tumor are still active in the primary cells. We used cells at low passage (passage 5) for the analysis. We have added this information in material and methods (line 420 f)

- How did the authors decide on which altered signaling pathways to treat? For example, vismodegib appeared to be effective in the in vitro screen, but there is no mention of why this drug was not used for the patient.

We selected the pathways to treat based on the biological relevance of the pathways, safety concerns and the availability of FDA released drugs. To date there is no approved Notch and WNT targeted therapy in the clinic and the development of life threatening toxicities associated with the targeting of these pathways still remain a concern [1] [2]. The SHH pathway can be targeted using vismodegib (released for basal cell carcinoma) and ATO (released for APL and with off-target effect on GLI). No inhibitors for IGF1R have been released so far, but it is known from clinical studies that the inhibition of the IGF pathway has an acceptable safety profile [3]. We previously showed that ceritinib can inhibit the IGF receptor IGF1R in a particular subtype of pediatric brain tumors at a concentration that is achievable in vivo [4]. We decided to start with the inhibition of the SHH pathway because the involvement of the SHH pathway in the biology of pediatric brain tumors is well established [5]. We didn’t use vismodegib to target the SHH pathway because in our in vitro model it reduced the cell proliferation to about 85% of the control, while the effect on the proliferation of ATO was much more impressive (about 16% of the control). Moreover, SMO inhibition causes permanent defects in bone structure in young mice and youg children, and its use has to be carefully considered with respect to long-term toxicities [6,7]. We have included these considerations in the results (line 215 ff and line 266 ff).

- Was the backbone chemotherapy used concurrently with targeted therapy the same or different as after the first recurrence? If different, why was it thought that additional targeted therapy was needed at the same time? The use of so many different agents makes it more difficult to tell the effective of the target therapy. Along these lines, it is not clear why the background chemotherapy was abandoned in favor of ceretinib monotherapy.

The backbone chemotherapy applied during the first recurrence included cyclophosphamide/vincristine and carboplatin/etoposide. We have included this information at lane 86. As backbone chemotherapy we incorporated systemic VA (vincristine and actinomycin-D) and VAd (adriamycin) blocks and intraventricular (via Ommaya) cytarabine and etoposide Gry. Arsenic trioxide synergizes with vincristine, doxorubicin and etoposide [8] [9]. Actinomycin-D was included to re-establish the tumor suppressive function of TP53 [10]. The intraventricular therapy was based on the MEMMAT protocol (NCT01356290) that has shown activity in different pediatric brain tumors [11].

By switching to ceritinib, because no synergisms have been described so far between ceritinib and chemotherapy, we decided to use the drugs as monotherapy. We have added this comment at line 274 ff.

- The imaging and text are not completely clear about which lesion was responsive to ceretinib monotherapy. If it is the cerebellar lesion that is thought to have responded, this response does not appear to be very significant. Is the FLAIR signal in the cerebellum entirely the tumor, or could it be edema associated with a very small lesion. Could it be that differences in the FLAIR signal are related to changes in edema instead of tumor? Also, the coronal slice is not quite the same pre- versus post-therapy, and I would be concerned that the slight difference in size of the cerebellar FLAIR signal could just be because of different locations are being looked at.

The larger right temporal lesion developed diffuse hemorrhagic transformation within one month under therapy with Ceritinib, which we consider to be a treatment effect. This effect is clearly shown on images A/B vs. D/E. The increased edema in FLAIR imaging (C vs. F) seems to be related to this hemorrhagic transformation.

With regard to the smaller cerebellar lesion we are sorry that we are not able to supply completely comparable images due to slight variations in protocols and angulation. However, this quite small, contrast enhancing lesion best seen in FLAIR (post contrast as shown in C and F) definitely has reduced contrast uptake and this is not a partial volume effect: we modified the figure to better show the decrease in size/contrast enhancement by adding magnified images.

Taken together, hemorrhagic transformation of the complete larger temporal lesion together with reduced contrast enhancement of the smaller lesion must be related to treatment effects of Ceritinib.

We have added these comments in the text (line 320 ff) and in Figure 7.

Ryeom, S.W. The cautionary tale of side effects of chronic Notch1 inhibition. J Clin Invest 2011, 121, 508-509, doi:10.1172/JCI45976. Kahn, M. Can we safely target the WNT pathway? Nat Rev Drug Discov 2014, 13, 513-532, doi:10.1038/nrd4233. Beckwith, H.; Yee, D. Minireview: Were the IGF Signaling Inhibitors All Bad? Mol Endocrinol 2015, 29, 1549-1557, doi:10.1210/me.2015-1157. Vewinger, N.; Huprich, S.; Seidmann, L.; Russo, A.; Alt, F.; Bender, H.; Sommer, C.; Samuel, D.; Lehmann, N.; Backes, N., et al. IGF1R Is a Potential New Therapeutic Target for HGNET-BCOR Brain Tumor Patients. International Journal of Molecular Sciences 2019, 20, 3027. Shahi, M.H.; Rey, J.A.; Castresana, J.S. The sonic hedgehog-GLI1 signaling pathway in brain tumor development. Expert Opin Ther Targets 2012, 16, 1227-1238, doi:10.1517/14728222.2012.720975. Kimura, H.; Ng, J.M.; Curran, T. Transient inhibition of the Hedgehog pathway in young mice causes permanent defects in bone structure. Cancer Cell 2008, 13, 249-260, doi:10.1016/j.ccr.2008.01.027. Robinson, G.W.; Kaste, S.C.; Chemaitilly, W.; Bowers, D.C.; Laughton, S.; Smith, A.; Gottardo, N.G.; Partap, S.; Bendel, A.; Wright, K.D., et al. Irreversible growth plate fusions in children with medulloblastoma treated with a targeted hedgehog pathway inhibitor. Oncotarget 2017, 8, 69295-69302, doi:10.18632/oncotarget.20619. Meister, M.T.; Boedicker, C.; Graab, U.; Hugle, M.; Hahn, H.; Klingebiel, T.; Fulda, S. Arsenic trioxide induces Noxa-dependent apoptosis in rhabdomyosarcoma cells and synergizes with antimicrotubule drugs. Cancer Lett 2016, 381, 287-295, doi:10.1016/j.canlet.2016.07.007. Boehme, K.A.; Nitsch, J.; Riester, R.; Handgretinger, R.; Schleicher, S.B.; Kluba, T.; Traub, F. Arsenic trioxide potentiates the effectiveness of etoposide in Ewing sarcomas. Int J Oncol 2016, 49, 2135-2146, doi:10.3892/ijo.2016.3700. Tzaridis, T.; Milde, T.; Pajtler, K.W.; Bender, S.; Jones, D.T.; Muller, S.; Wittmann, A.; Schlotter, M.; Kulozik, A.E.; Lichter, P., et al. Low-dose Actinomycin-D treatment re-establishes the tumoursuppressive function of P53 in RELA-positive ependymoma. Oncotarget 2016, 7, 61860-61873, doi:10.18632/oncotarget.11452. Slavc, I.; Peyrl, A.; Chocholous, M.; Reisinger, D.; Mayr, L.; Azizi, A.; Dieckmann, K.; Haberler, C.; Czech, T. MBCL-27. RESPONSE OF RECURRENT MALIGNANT CHILDHOOD CNS TUMORS TO A MEMMAT BASED METRONOMIC ANTIANGIOGENIC COMBINATION THERAPY VARIES DEPENDENT ON TUMOR TYPE: EXPERIENCE IN 71 PATIENTS. Neuro-Oncology 2018, 20, i122-i122, doi:10.1093/neuonc/noy059.423.

Reviewer 3 Report

The authors have provided a detailed workup and evaluation, treatment of a single patient with Ceritinib for a neuroepithelial brain tumor with likely mechanism of regressing IGF tumors.

Their histopathological details, immunohistochemistry and molecular marker detail along with gene analysis is very detailed and leaves no stone unturned when looking at the molecular basis of these tumors. Their DNA and RNA sequencing details are well described as well, with good images to match their descriptions. They have also done a didactic literature review and described their material methods section in great details.

Whether this one case really provides substantial data need confirmation with multicenter prospective RCTs to answer the question whether ceritinib definitively works against IGFR in these tumors and would recommend the authors look towards collaborating or carrying out this in the future

Author Response

We thank the reviewer for the positive feedback. We have added a statement on study limitations (from line 405)

Round 2

Reviewer 2 Report

In the revised manuscript, the authors have improved the level of detail of the clinical treatment paradigms and scientific experiments. Moreover, they better describe their rationale for choices made choosing specific pathways to treat/drug agents to use.

It is still not quite clear to me why ceretinib was initially used with other chemotherapeutic agents but then used as monotherapy. The authors note in their Response that there is no previous report of synergism between ceretinib and other chemotherapies. If this is the case, why did the authors use the two together to start?

I also have a problem with the authors' interpretation that tumor hemorrhage represents a  response to ceretinib. How can the authors tell if the hemorrhage was related to ceretinib effect or a spontaneous event. I would also say that if the primary treatment response of  a tumor to ceretinib was hemorrhage causing more mass effect and edema, this would be problematic. I think that the conclusion that tumor hemorrhage represents a treatment response to ceretinib needs to be significantly tempered. I do not have a problem with the authors concluding that there is a treatment response to ceretinib based on decreased enhancement of the cerebellar lesion, but the lesion is small to start, and so the effect does not seem to be particularly impressive. 

Author Response

It is still not quite clear to me why ceretinib was initially used with other chemotherapeutic agents but then used as monotherapy. The authors note in their Response that there is no previous report of synergism between ceretinib and other chemotherapies. If this is the case, why did the authors use the two together to start?

Due to the high malignancy of the disease we initially decided to maintain the chemiotherapy backbone in parallel to ceritinib for the first four weeks. Furthermore it was unknown, if the efficacy of the ceritinib monotherapy, successfully used to target ALK, was equivalent in the context of an off targeting IGF inhibition. The combination of IGF inhibition and chemotherapy is supported by clinical data showing that the inhibition of IGF signaling can enhance the effects of chemotherapy [41]. The combined therapy was maintained until the detection of the new cerebellar lesion (blue star in Figure 6). At this point, we decided to apply ceritinib alone. The rationale was at one hand that no further benefit was observed and at the other hand, so far no clinically proven interaction of ceritinib with chemotherapy was known. Ceritinib has been shown to be effective as monotherapy in brain metastasis of lung cancer (with a median time to intracranial response of 6.1 weeks [42]). Additionally, we intended to improve the quality of life of the patient avoiding further hospitalization. We have added these comments in the text at line 310 ff.

I also have a problem with the authors' interpretation that tumor hemorrhage represents a  response to ceretinib. How can the authors tell if the hemorrhage was related to ceretinib effect or a spontaneous event. I would also say that if the primary treatment response of  a tumor to ceretinib was hemorrhage causing more mass effect and edema, this would be problematic. I think that the conclusion that tumor hemorrhage represents a treatment response to ceretinib needs to be significantly tempered.

 I do not have a problem with the authors concluding that there is a treatment response to ceretinib based on decreased enhancement of the cerebellar lesion, but the lesion is small to start, and so the effect does not seem to be particularly impressive. 

We thank the reviewer for this hint. The conclusion that the hemorrhagic transformation of the tumor is a therapeutic effect relates to the histopathological detection of necrotic tumor area in the OP preparation of this formation.

We changed the sentence to (line 330 f):

The larger right temporal lesion developed diffuse hemorrhagic transformation under therapy with ceritinib (Figure 7, A/B vs. D/E) and was removed surgically 10 days later. Histopathology demonstrated near complete necrosis as well as bleeding with only scattered areas of viable cells. Bleeding and necrosis has been previously described in association with response to IGF1R inhibitors in relapsed malignant astrocytoma and squamous non-small cell lung carcinoma [43,44]. It remains to be clarified if an off-target inhibition by ceritinib is comparable to an inhibition with specific IGF inhibitors in term of adverse events like hemorrhage.

41-Macaulay, V.M.; Middleton, M.R.; Protheroe, A.S.; Tolcher, A.; Dieras, V.; Sessa, C.; Bahleda, R.; Blay, J.Y.; LoRusso, P.; Mery-Mignard, D., et al. Phase I study of humanized monoclonal antibody AVE1642 directed against the type 1 insulin-like growth factor receptor (IGF-1R), administered in combination with anticancer therapies to patients with advanced solid tumors. Ann Oncol 2013, 24, 784-791, doi:10.1093/annonc/mds511.

42-Kim, D.W.; Mehra, R.; Tan, D.S.; Felip, E.; Chow, L.Q.; Camidge, D.R.; Vansteenkiste, J.; Sharma, S.; De Pas, T.; Riely, G.J., et al. Activity and safety of ceritinib in patients with ALK-rearranged non-small-cell lung cancer (ASCEND-1): updated results from the multicentre, open-label, phase 1 trial. Lancet Oncol 2016, 17, 452-463, doi:10.1016/S1470-2045(15)00614-2.

43-Aiken, R.; Axelson, M.; Harmenberg, J.; Klockare, M.; Larsson, O.; Wassberg, C. Phase I clinical trial of AXL1717 for treatment of relapsed malignant astrocytomas: analysis of dose and response. Oncotarget 2017, 8, 81501-81510, doi:10.18632/oncotarget.20662.

44-Ekman, S.; Frodin, J.E.; Harmenberg, J.; Bergman, A.; Hedlund, A.; Dahg, P.; Alvfors, C.; Stahl, B.; Bergstrom, S.; Bergqvist, M. Clinical Phase I study with an Insulin-like Growth Factor-1 receptor inhibitor: experiences in patients with squamous non-small cell lung carcinoma. Acta Oncol 2011, 50, 441-447, doi:10.3109/0284186X.2010.499370.